# Leveraging Class Similarity for Enhanced Conformal Prediction

## Abstract

Conformal Prediction (CP) has emerged as a powerful statistical framework for high-stakes classification applications. Instead of predicting a single class, CP generates a prediction set, guaranteed to include the true label with a pre-specified probability. In setups where the classes can be partitioned into semantic groups, e.g., diseases that require similar treatment, users can benefit from prediction sets that are not only small on average, but also contain a small number of semantically different groups. This paper begins by addressing this problem and then extends far beyond it. First, given a class partition, we propose augmenting the CP score function with a term that penalizes predictions with "out-of-group" errors. We theoretically analyze this strategy and prove its advantages for group-related metrics. Surprisingly, we show mathematically that, for common class partitions, it can also reduce the average set size of *any* CP score function. Our analysis reveals the class similarity factors behind this improvement and motivates us to propose a model-specific variant, which *does not require* any human semantic partition and can further reduce the prediction set size. Finally, we present an extensive empirical study, encompassing prominent CP methods, multiple models, and several datasets, which demonstrates that our class-similarity-based approach consistently enhances CP methods, making it a widely applicable tool in the CP toolbox.

## 1 Introduction

Conformal Prediction (CP) (Vovk et al., 1999; 2005) has emerged as a powerful statistical framework for high-stakes classification applications, such as medical diagnoses (Lambert et al., 2024) and autonomous vehicle decision-making (Lindemann et al., 2024). Rather than predicting a single label, the CP framework outputs a set of candidate labels with a formal guarantee of marginal coverage: under exchangeability of the calibration and test samples, the prediction set will include the correct label with a user-specified probability. This property makes CP particularly valuable in safety-critical domains, where missing the correct label can have severe consequences. A key metric for comparing different CP methods is the average size of their prediction sets, commonly referred to in the literature as *efficiency* (Sadinle et al., 2019; Angelopoulos et al., 2021; Huang et al., 2024; Dabah & Tirer, 2025).

In many practical scenarios, classes can be grouped into semantic categories, and users can benefit from prediction sets that are not only small on average but also contain semantically similar classes. For example, consider an autonomous driving system that relies on a classifier, which outputs sets of likely scene interpretations from its camera feed. Even if the classifier achieves high coverage (e.g., its prediction sets contain the correct scene label in at least 90% of cases), the prediction sets can still include mutually incompatible situations, such as {clear road, pedestrian crossing, oncoming train}. Although the true scene label is usually present, the wide variance in the associated outcomes makes it difficult to select a safe action, such as whether to brake, accelerate, or steer. In such scenarios, the system would be unsafe to base its decisions on the classifier, despite its high coverage. A similar example arises in healthcare: a disease classifier that mostly outputs prediction sets with diseases that require similar treatment, is expected to be more practically useful than one that does not, given the same coverage rate and efficiency. These examples highlight a critical limitation of current CP approaches: they ensure that the true label is included but fail to account for the semantic coherence of labels within the prediction set.

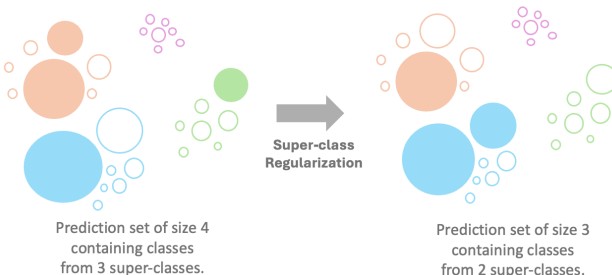

Figure 1: Illustration of prediction sets for an example before and after applying our proposed regularization. Each circle corresponds to a class, with colors indicating superclasses. Filled circles denote classes included in the prediction set, and circle size reflects the softmax value. In this example, the prediction set size decreases from 4 to 3, and the number of superclasses represented decreases from 3 to 2. We show that our regularization typically reduces the prediction set size.

This paper addresses this gap and extends beyond it. First, assuming a known partition of classes into groups, we propose augmenting the CP score function with a binary regularization term that penalizes predictions with "out-of-group" errors. We provide a theoretical analysis of this strategy and prove that it reduces the expected number of unique groups appearing in a prediction set. Interestingly, our theory also reveals a surprising property: for common class partitions, applying this penalty can simultaneously decrease the average prediction set size, regardless of the underlying CP score function. Our analysis further identifies the class similarity factors behind this improvement. Motivated by these insights, we extend our approach and propose a model-specific variant, which does not require any human semantic partition. Specifically, we construct a class similarity matrix from the classifier's embedding vectors, leveraging the model's own perception of similar classes. This enables regularization that can further reduce the prediction set size and does not require any external knowledge of class groups, making it applicable for general datasets. Finally, we present an extensive empirical study evaluating the performance of both variants, the one that uses a known, Model-Agnostic (MA), class partition, and the one that relies on Model-Specific (MS) class similarities. Our experiments encompass multiple datasets, models, and prominent CP score functions (LAC (Sadinle et al., 2019), RAPS (Angelopoulos et al., 2021), and SAPS (Huang et al., 2024)). We show that our class-similarity-based approach consistently enhances each of these diverse CP methods, providing a flexible and widely applicable tool for improving both the coherence and efficiency of prediction sets.

**Our contributions.** Our main contributions can be summarized as follows.

- We propose a score regularization approach, independent of the original CP score function, which improves both the semantic coherence of prediction sets and their average size.
- We provide a theoretical analysis of the proposed binary penalty, proving its effectiveness in reducing the expected number of unique groups in the prediction set, as well as decreasing the average set size.
- We introduce a model-specific variant for applying the regularization, which further reduces the prediction set size and does not require any known class structure.
- We conduct an extensive empirical evaluation across multiple dataset-model pairs, demonstrating that both our model-agnostic and model-specific regularizations consistently enhance prominent CP methods, outperforming their original versions in both semantic coherence and size of their prediction sets.

## 2 RELATED WORK

**Clustered and group-conditional coverage CP.** (Vovk, 2012; Ding et al., 2023; Bairaktari et al., 2025) aim to improve coverage across heterogeneous label groups. These methods typically apply the CP procedure separately within each group, or cluster classes based on model scores, yielding prediction sets that have better group conditional coverage but larger prediction set size than their baselines.

**Hierarchical and structured CP.** Other works incorporate a known label hierarchy, such as a directed acyclic graph, into the conformal prediction framework. (Hengst et al., 2025) and (Zhang et al., 2024) share the objective of controlling the specificity of prediction sets (e.g., the number of leaf labels) alongside efficiency. (Mortier et al., 2025) introduce the notion of representation complexity, defined as the minimum number of nodes whose descendants cover the prediction set, and study its trade-off with efficiency.

**Hierarchical selective classification.** When a hierarchical structure of labels is available, with classes located at the leaves, Goren et al. (2024) extend the conformal prediction framework to achieve hierarchical selective coverage. Their approach identifies a predicted node by starting from the predicted class (which is a leaf) and ascending the hierarchy until a conformal threshold is met, in a manner similar to the APS procedure (Romano et al., 2020). This method focuses on controlling the trade-off between predictive accuracy and the specificity of the hierarchical prediction.

**Summary.** To our knowledge, no prior work directly addresses the objective of improving semantic coherence within prediction sets without compromising CP efficiency, let alone leveraging class structure to improve efficiency. Existing works in clustered, group-conditional, and hierarchical CP focus on coverage within known groups or on relatively uncommon notions of structure. Importantly, these approaches typically yield prediction sets that are larger than the baselines. In contrast, our approach reduces the number of semantically distinct groups, decreases the average set size, and is applicable across datasets and CP score functions. Moreover, our model-specific approach does not require any prior knowledge of class structure.

## 3 PRELIMINARIES ON CONFORMAL PREDICTION

Let us present notations that are used in the paper, followed by some preliminaries on CP. We consider a $C$-classes classification task of the data $(X, Y)$ distributed on $\mathcal{X} \times [C]$, where $[C] := \{1, \ldots, C\}$. The task is addressed by a classifier model (e.g., a trained deep neural network) that for each input sample $x \in \mathcal{X}$ produces a post-softmax probability vector $\hat{\pi}(x) \in \mathbb{R}^C$. The predicted class is given by $\hat{y}(x) = \operatorname{argmax}_i \hat{\pi}_i(x)$.

Conformal Prediction (CP) is a methodology for reliable classification, independent of the data distribution. Given a black-box classifier, predefined $\alpha \in (0, 1)$, and a sample $X$, it generates a *prediction set* of classes, $\mathcal{C}(X)$, such that $Y \in \mathcal{C}_\alpha(X)$ with probability $1 - \alpha$, where $Y$ is the true class associated with $X$ (Vovk et al., 1999; 2005; Papadopoulos et al., 2002). The decision rule is based on a calibration set of labeled samples $\{x_i, y_i\}_{i=1}^n$. The only assumption in CP is that the random variables associated with the calibration set and the test samples are exchangeable (e.g., the samples are i.i.d.).

Let us state the general process of conformal prediction given the calibration set $\{x_i, y_i\}_{i=1}^n$ and its deployment for a new (test) sample $x_{n+1}$ (for which $y_{n+1}$ is unknown), as presented in (Angelopoulos & Bates, 2021):

1. Define a heuristic score function $s(x, y) \in \mathbb{R}$ based on some output of the model. A higher score should encode a lower level of agreement between $x$ and $y$.

2. Compute $\hat{q}$ as the $\lceil (n+1)(1-\alpha) \rceil / n$ quantile of the scores $\{s(x_1, y_1), \ldots, s(x_n, y_n)\}$.

3. At deployment, create the prediction set of a test sample as $\mathcal{C}(x_{n+1}) = \{y : s(x_{n+1}, y) \leq \hat{q}\}$.

CP methods possess the following coverage guarantee.

**Theorem 3.1** (Theorem 1 in (Angelopoulos & Bates, 2021)). *Suppose that $\{(X_i, Y_i)\}_{i=1}^n$ and $(X_{n+1}, Y_{n+1})$ are i.i.d., and define $\hat{q}$ as in step 2 above and $\mathcal{C}_\alpha(X_{n+1})$ as in step 3 above. Then, $\mathbb{P}(Y_{n+1} \in \mathcal{C}(X_{n+1})) \geq 1 - \alpha$.*

The proof of this result is based on (Vovk et al., 1999). A proof of an upper bound of $1 - \alpha + 1/(n+1)$ also exists. Note that the coverage is marginal: the probability is taken over the entire distribution of $(X, Y)$ and there is no guarantee per value of $X_{n+1}$.

Different CP methods typically differ by their choice of score function $s(x, y)$, and a key property that they are judged according to is their average prediction set size, $\mathbb{E}[|\mathcal{C}(X)|]$, often refers to as *efficiency*.

## 4 ENHANCING CP USING CLASS SIMILARITY

In this section, we explore the properties of CP when utilizing a known partition of the $C$ classes into $G$ groups. Let $g : [C] \to [G]$ denote the map of classes to groups. Namely, $g(y) \in [G]$ is the index of the group that contains class $y \in [C]$.

As discussed in Section 1, we assume that the groups are superclasses with some semantical meaning. For example, the classes may be cities and the superclasses are geographical location, or the classes are types of diseases and the superclasses group those that require a similar treatment. In this case, it is reasonable for a user to prefer $\mathcal{C}(X)$ whose classes belong only to a few groups, or ideally just to the group of the true label $Y$.

Motivated by the above, let us set a "distance function" between classes based on their groups. Specifically, we consider the binary penalty function given by:

$$d(y, y') := \mathbb{I}\{g(y) \neq g(y')\}, \tag{1}$$

where $\mathbb{I}\{\cdot\}$ is the indicator function. That is, $d(y, y') = 0$ if $y$ and $y'$ belong to the same group, and otherwise $d(y, y') = 1$. For brevity, we omit the explicit dependence of $d(y, y')$ on $g$.

Given a sample $x$, all common CP methods preserve the ranking of the softmax vector $\hat{\pi}(x)$ and, in particular, include the estimated class $\hat{y}(x)$ in the prediction set before including any other class. Therefore, to reduce the number of groups in $\mathcal{C}(X)$ we propose to penalize a given score function $s(x, y)$ by $d(y, \hat{y}(x))$:

$$s_\lambda(x, y) := s(x, y) + \lambda d(y, \hat{y}(x)), \tag{2}$$

where $\lambda > 0$ is a parameter. In words, the score of a candidate $y$ that is "semantically far" from $\hat{y}(x)$ is penalized by a value of $\lambda$.

We turn to theoretically explore the properties of CP with $s_\lambda(x, y)$. Let us denote by $\hat{q}_\lambda$ and $\mathcal{C}_\lambda(x)$ the CP threshold and prediction set when using $\lambda > 0$.

**The coverage property is maintained.** This follows directly from Theorem 3.1, as $s_\lambda$ is a valid score that preserves the exchangeability of the calibration and test samples.

**The number of out-of-group labels in the prediction set cannot increase.** To show that adding the penalty term to the score cannot increase the number of classes in $\mathcal{C}_\lambda(x)$ whose group is not $g(\hat{y}(x))$, we first establish the following lemma on the relation between $\hat{q}_\lambda$ and $\hat{q}$.

**Lemma 4.1.** *We have $\hat{q} \leq \hat{q}_\lambda \leq \hat{q} + \lambda$.*

*Proof.* For any $(x_i, y_i)$ in the calibration set we have $s(x_i, y_i) \leq s_\lambda(x_i, y_i) \leq s(x_i, y_i) + \lambda$. The $(1 - \alpha)$ empirical quantile for $\{s(x_i, y_i)\}$ is $\hat{q}$ and for $\{s(x_i, y_i) + \lambda\}$ is $\hat{q} + \lambda$. Therefore the $(1 - \alpha)$ empirical quantile for $\{s_\lambda(x_i, y_i)\}$ is in $[\hat{q}, \hat{q} + \lambda]$. $\square$

In fact, a slightly more detailed analysis can be show that $\hat{q}_\lambda$ is non-decreasing with $\lambda$, and $(\hat{q}_\lambda - \lambda)$ is non-increasing with $\lambda$, but the above result suffices for our analysis. We now present our result on the inclusion of out-of-group labels in the prediction set.

**Proposition 4.2.** *Let $\mathcal{Y}_1(x) := \{y : d(y, \hat{y}(x)) \neq 0\}$. For any $x$ and $\lambda > 0$ we have*

$$\mathcal{C}_\lambda(x) \cap \mathcal{Y}_1(x) \subseteq \mathcal{C}(x) \cap \mathcal{Y}_1(x).$$

*Proof.* For any $y \in \mathcal{Y}_1(x)$, we have $d(y, \hat{y}(x)) = 1$, so the inclusion in $\mathcal{C}_\lambda(x)$ implies satisfying $s(x, y) + \lambda \leq \hat{q}_\lambda \leq \hat{q} + \lambda$, where the second inequality follows from Lemma 4.1. Therefore, $s(x, y) \leq \hat{q}$, which implies $y \in \mathcal{C}(x)$. $\square$

The proposition shows that the penalization cannot add any "far"/group-mismatched labels (w.r.t. $\hat{y}(x)$) that weren't already in the unpenalized CP; it can only remove them. This property naturally translates to decreasing the distance-weighted size. Specifically, define $S_\lambda(x) := \sum_{y=1}^{C} d(y, \hat{y}(x)) \mathbb{I}\{y \in \mathcal{C}_\lambda(x)\}$. We have $S_\lambda(x) \leq S_0(x)$ for any $x$, and thus also $\mathbb{E}[S_\lambda(X)] \leq \mathbb{E}[S_0(X)]$. Similarly, eliminating the pathological case of empty $\mathcal{C}(x)$, the number of groups cannot increase, as shown in the following corollary.

**Corollary 4.3.** *Let $\mathcal{G}_\lambda(x)$ and $\mathcal{G}(x)$ denote the groups represented in $\mathcal{C}_\lambda(x)$ and $\mathcal{C}(x)$, respectively. For any $x$ such that $\hat{y}(x) \in \mathcal{C}(x)$ and $\lambda > 0$ we have $G_\lambda(x) \subseteq G(x)$.*

*Proof.* Formally, $\mathcal{G}_\lambda(x) = \{g(y) : y \in \mathcal{C}_\lambda(x)\}$ and $\mathcal{G}(x) = \{g(y) : y \in \mathcal{C}(x)\}$. We already have in Proposition 4.2 that any $y$ with $g(y) \neq g(\hat{y}(x))$ cannot be added to $\mathcal{C}_\lambda(x)$. The assumption that $\hat{y}(x) \in \mathcal{C}(x)$ eliminates the pathological case that $s_\lambda(x, \hat{y}(x)) \leq \hat{q}_\lambda$ but $s(x, \hat{y}(x)) > \hat{q}$. So, $g(\hat{y}(x))$ is already in both $\mathcal{G}_\lambda(x)$ and $\mathcal{G}(x)$. $\qquad\square$

Since the corollary holds for any $x$, it reflects the relation $\mathbb{E}[|\mathcal{G}_\lambda(X)|] \leq \mathbb{E}[|\mathcal{G}(X)|]$. This theory supports the empirical observation (in Section 6) that the empirical expectations obey $\hat{\mathbb{E}}[|\mathcal{G}_\lambda(X)|] < \hat{\mathbb{E}}[|\mathcal{G}(X)|]$, with a substantial margin.

**Surprising behavior: The average prediction set size also decreases in practice.** As will be shown in our experiments, with well-tuned $\lambda$ (small enough), we observe $\hat{\mathbb{E}}[|\mathcal{C}_\lambda(X)|] < \hat{\mathbb{E}}[|\mathcal{C}(X)|]$, in benchmark settings, even though the penalty does not imply it directly. Actually, while we reached a guarantee for decreasing the number of "out-of-group" labels, potentially, there can be an increase in "in-group" labels, since $\hat{q} \leq \hat{q}_\lambda$ but the score of $y$ from the same group of $\hat{y}(x)$ remains the same.

We turn to establish a theory for reduction in the average prediction set size for small enough $\lambda$. To this end, let us start by some definitions.

**Definition 4.4.** *Given a sample $x$, we have the following definitions:*

1. *"In-group" classes: $\mathcal{Y}_0(x) := \{y : d(y, \hat{y}(x)) = 0\}$ and $n_0(x) := |\mathcal{Y}_0(x)|$.*

2. *"Out-of-group" classes: $\mathcal{Y}_1(x) := \{y : d(y, \hat{y}(x)) \neq 0\}$ and $n_1(x) := |\mathcal{Y}_1(x)|$.*

3. *Per-$x$ conditional quasi-CDF:[1] for $z \in \{0, 1\}$, $\hat{F}_z^x(t) := \dfrac{1}{n_z(x)} \displaystyle\sum_{y \in \mathcal{Y}_z(x)} \mathbb{I}\{s(x, y) \leq t\}$.*

*We also make the following definitions related to the marginal distribution:*

4. *Average number of "in-group" classes: $\overline{n}_0 := \mathbb{E}[n_0(X)]$.*

5. *Average number of "out-of-group" classes: $\overline{n}_1 := \mathbb{E}[n_1(X)]$.*

6. *Probability of "in-group" true label: $p_0 = \mathbb{P}(Y \in \mathcal{Y}_0(X))$.*

7. *Probability of "out-of-group" true label: $p_1 = \mathbb{P}(Y \in \mathcal{Y}_1(X)) = 1 - p_0$.*

8. *Conditional CDFs: for $z \in \{0, 1\}$, $F_z(t) := \mathbb{P}(s(X, Y) \leq t | Y \in \mathcal{Y}_z(X))$.*

Next, let us state the assumptions that will be used in our theorem.

**Assumption 1.** *For small $\lambda \geq 0$, the prediction set $\mathcal{C}_\lambda(X)$ is based on the statistical quantile $q_\lambda$ of the CDF of $s_\lambda(X, Y)$.*

**Assumption 2.** *For $z \in \{0, 1\}$, the CDF $F_z(t)$ is absolutely continuous, so $f_z(t) = F_z'(t)$ is well-defined.*

**Assumption 3.** *For $z \in \{0, 1\}$, the "size-biased" quasi-CDF $\tilde{F}_z(t) := \dfrac{1}{\overline{n}_z} \mathbb{E}[n_z(X)\hat{F}_z^X(t)]$ is absolutely continuous, so $\tilde{f}_z(t) = \tilde{F}_z'(t)$ is well-defined.*

Assumptions 1-3 are required for making the analysis tractable, ensuring that $\mathbb{E}[|\mathcal{C}_\lambda(X)|]$ is differentiable with respect to $\lambda$, and sparing cumbersome analysis of the effect of finite calibration sets on inclusion of a label in the predictions sets. Note that Assumption 1 essentially reflects having a large calibration set. Now we present a theorem that characterizes the effect of the penalty with small $\lambda$ on the efficiency.

---

[1] We name the object $\hat{F}_z^x(t)$ "quasi-CDF" because it is not based on any random variable (such as $Y|X = x$) or its realization, but rather on the deterministic set $\mathcal{Y}_z(x)$.

**Theorem 4.5.** *Consider Definition 4.4. Under Assumptions 1-3, we have*

$$\text{sign}\left(\frac{\mathrm{d}}{\mathrm{d}\lambda}\mathbb{E}[|\mathcal{C}_\lambda(X)|]\big|_{\lambda=0}\right) = \text{sign}\left(ap_1\overline{n}_0 - bp_0\overline{n}_1\right), \tag{3}$$

*where $a := \tilde{f}_0(q_0)f_1(q_0)$ and $b := \tilde{f}_1(q_0)f_0(q_0)$.*

*Proof.* See Appendix A.1. The proof sketch is as follows. We establish an expression for the score function CDF, $F_\lambda(t) := \mathbb{P}(s_\lambda(X,Y) \leq t)$ in terms of the conditional CDFs, $F_0(t)$ and $F_1(t)$. By Assumption 1, we have $F_\lambda(q_\lambda) = 1 - \alpha$, on which we apply implicit differentiation and establish an expression for $\frac{dq_\lambda}{d\lambda}$. Based on the definitions we express $\mathbb{E}[|\mathcal{C}_\lambda(X)|]$ using the "size-biased" quasi-CDF. Differentiating it and substituting $\frac{dq_\lambda}{d\lambda}$ at $\lambda = 0$ leads to the advertised result.

$\square$

**Discussion.** Let us start by assuming that $a \approx b$. In this case, the theorem shows that the sign of $\frac{\mathrm{d}}{\mathrm{d}\lambda}\mathbb{E}[|\mathcal{C}_\lambda(X)|]\big|_{\lambda=0}$ (where a negative value means that $\lambda \approx 0_+$ reduces $\mathbb{E}[|\mathcal{C}_\lambda(X)|]$) equals the sign of the difference between:

- $p_1 \times \overline{n}_0$: The probability of having the true label out of the group of the predicted class $\times$ the average number of classes in the group of the predicted class.

- $p_0 \times \overline{n}_1$: The probability of having the true label in the group of the predicted class $\times$ the average number of classes out of the group of the predicted class.

We can expect that $p_1\overline{n}_0 \ll p_0\overline{n}_1$ in most practical cases, since typically the number of classes in a group is much smaller than outside a group, i.e., $\overline{n}_0 \ll \overline{n}_1$, and modern classifiers are quite powerful so $p_0$ is not small. Therefore, if $a \approx b$ or even if $b$ is not much smaller than $a$, then $\text{sign}(ap_1\overline{n}_0 - bp_0\overline{n}_1) < 0$. By Theorem 4.5, this implies that penalizing the score with small $\lambda$ will reduce $\mathbb{E}[|\mathcal{C}_\lambda(X)|]$.

Let us discuss the relation between $a$ and $b$. For simplification, assume that the $C$ classes are partitioned to $G$ groups of equal size $K$. In this case, we have constants $n_0(X) = K$ and $n_1(X) = (G-1)K$. So, $\overline{n}_0 = K$ and $\overline{n}_1 = (G-1)K$, as well. Recalling the definition of $\tilde{F}_z(t)$ in Assumption 3, we have

$$\tilde{F}_z(t) = \mathbb{E}[\hat{F}_z^X(t)] = \frac{1}{\overline{n}_z}\mathbb{E}\left[\sum_{y\in\mathcal{Y}_z(X)}\mathbb{I}\{s(X,y)\leq t\}\right].$$

Define $p_z(x) := \mathbb{P}(Y \in \mathcal{Y}_z(x)|X = x)$ and $F_z^x(t) := \mathbb{P}(s(x,Y) \leq t|Y \in \mathcal{Y}_z(x), X = x)$. Observe that

$$F_z^x(t) = \frac{\mathbb{P}(s(x,Y) \leq t, Y \in \mathcal{Y}_z(x)|X = x)}{\mathbb{P}(Y \in \mathcal{Y}_z(x)|X = x)}$$

$$= \frac{\sum_{y\in\mathcal{Y}_z(x)}\mathbb{P}(Y = y|X = x)\mathbb{I}\{s(x,y) \leq t\}}{p_z(x)}.$$

Using the relation (see derivation in Appendix A.2):

$$F_z(t) = \frac{1}{p_z}\mathbb{E}[p_z(X)F_z^X(t)] \tag{4}$$

and substituting in it the expression derived above for $F_z^x(t)$, we get

$$F_z(t) = \frac{1}{p_z}\mathbb{E}[p_z(X)F_z^X(t)] = \frac{1}{p_z}\mathbb{E}\left[\sum_{y\in\mathcal{Y}_z(X)}\mathbb{P}(Y = y|X)\mathbb{I}\{s(X,y) \leq t\}\right].$$

Thus, if, per $X = x$, the labels $Y \in \mathcal{Y}_z(x)$ are distributed uniformly, then the factor that multiplies $\mathbb{I}\{s(X,y) \leq t\}$ is constant, and therefore $\tilde{F}_z(t) = F_z(t)$. This gives exact $a = b$ (recall their definitions in Theorem 4.5), so the above arguments hold for having $\text{sign}(ap_1\overline{n}_0 - bp_0\overline{n}_1) < 0$. The fact that the theory includes integration over $X$ and considers the densities only at $q_0$, together with the empirical fact that $p_1\overline{n}_0 \ll p_0\overline{n}_1$, teach us that there are many cases where $\text{sign}(ap_1\overline{n}_0 - bp_0\overline{n}_1) < 0$ even for non uniform conditional label distributions.

## 5 Extension to Model-Specific Class Similarity

The original motivation for the penalized score in equation 2 came from considering the potential preference of the user to reduce the number of "semantically far" classes in the prediction set. However, Theorem 4.5 reveals that, perhaps surprisingly, the proposed penalty has a beneficial effect on the prediction set size for any score function, provided that the penalty parameter $\lambda$ is sufficiently small and $p_1 \overline{n}_0 \ll p_0 \overline{n}_1$ (omitting the effect of $a$ and $b$ in equation 3).

This result actually tells us that, in terms of efficiency, we can gain more from partitions into groups that are as small as possible (low $\overline{n}_0$ and high $\overline{n}_1$), as long as the probability of making out-of-group mistakes ($p_1 = 1 - p_0$) is kept low. Nothing in this result requires a human-related semantic similarity between classes within a group. This motivates us to propose a *model-specific* extension of the method. Specifically, given a pretrained classifier, we suggest basing the penalty on the class similarity *perceived by the model*. An important advantage of this extension, which focuses on boosting efficiency rather than group-related metrics, is that it eliminates the need for a human-made semantic partition, which may not be available for some datasets.

For a given classifier, the proposed extension requires computing a $C \times C$ class similarity matrix, which we denote by $M$ (recall that $C$ denotes the number of classes). The $(c, c')$ entry in $M$ should reflect the similarity between class $c$ and class $c'$, as perceived by the model. Similarity metrics are typically continuous, e.g., inner products and kernels. Binarization of such metrics will require tuning a threshold parameter. Hence, we propose to diverge slightly from the method in Section 4 by allowing the similarity metric to be "soft", which also adds more flexibility to the method. For $M_{c,c'} \in \mathbb{R}$, upper bounded by 1 as the maximum level of similarity, we define the soft model-specific penalty function:

$$d^{MS}(y, y') := 1 - M_{y,y'}. \tag{5}$$

Substituting this penalty in equation 2 in lieu of the model-agnostic $d$, gives

$$s^{MS}_\lambda(x, y) := s(x, y) + \lambda d^{MS}(y, \hat{y}(x)), \tag{6}$$

where $\lambda > 0$ is a parameter.

**Determining model-specific class similarity.** There exist multiple potential strategies for constructing a class similarity matrix $M$ given a model. Here, we propose one that consistently improves the efficiency results in our experiments. Future research may attempt to optimize this choice. We assume access to the labeled training samples. The last layer of a deep neural network-based classifier $f(x) \in \mathbb{R}^C$, before the softmax operation, can be typically expressed as: $f(x) = W h_\theta(x) + b$, where $h_\theta(\cdot) : \mathcal{X} \to \mathbb{R}^p$ (with $p \geq C$) is the deepest feature mapping that is composed of all the hidden layers (with learnable parameters $\theta$), and $W \in \mathbb{R}^{C \times p}$ and $b \in \mathbb{R}^C$ are the weights and bias of the last classification layer. We determine the class similarity according to a similarity function (or kernel) between the means of different classes in the deepest feature space. This strategy is motivated by recent work on the neural collapse phenomenon (Papyan et al., 2020), where the within-class samples of well-trained classifiers concentrate around their class mean in feature space, while inter-class means are well separated yet empirically still preserve relations that generalize to test data (Tirer et al., 2023; Yang et al., 2023). Therefore, examining the relation between class means in feature space yields small effective groups without compromising on group-wise accuracy.

Denote by $\{x_{c,i}\}$, $i \in [n_c]$, the training samples associated with class $c \in [C]$. Compute the class means and the global mean of the features:

$$\overline{h}_c = \frac{1}{n_c} \sum_{i=1}^{n_c} h_\theta(x_{c,i}), \quad c \in [C]. \qquad \overline{h}_G = \frac{1}{C} \sum_{c=1}^{C} \overline{h}_c. \tag{7}$$

We then set the entry $M_{c,c'}$ in the class similarity matrix $M$ using the cosine similarity of the centered class means: $M_{c,c'} = \dfrac{\langle \overline{h}_c - \overline{h}_G, \overline{h}_{c'} - \overline{h}_G \rangle}{\|\overline{h}_c - \overline{h}_G\| \|\overline{h}_{c'} - \overline{h}_G\|}.$

## 6 Experiments

**Datasets and models.** We conduct experiments on three image classification benchmarks: CIFAR-100 (Krizhevsky et al., 2009), Living-17 from the BREEDS suite (Santurkar et al., 2020), and Mini-

ImageNet (Vinyals et al., 2016), a subset of ImageNet (Deng et al., 2009) with 100 classes. Note that CIFAR-100 and Living-17 have official semantic superclass structures, i.e., partitions of the classes into coarse classes. Specifically, CIFAR-100 has 20 superclasses (e.g., aquatic mammals, fish, flowers, etc.), where each one groups 5 classes (e.g., beaver, dolphin, otter, seal, and whale are grouped under aquatic mammals). Similarly, Living-17 has 17 superclasses, where each one groups 4 classes. We use ResNet50 (He et al., 2016) as the classifier model. For CIFAR-10 we use ResNet34 as well. Details on the training of the models are provided in Appendix B.1. We split the validation sets of the datasets to 20% calibration and 80% test.

**CP score functions.** We consider three prominent conformal score functions $s(x, y)$: (1) LAC (Sadinle et al., 2019), defined as one minus the classifier's softmax value at index $y$; (2) RAPS (Angelopoulos et al., 2021), based on cumulating softmax entries up to the rank of $y$, like APS (Romano et al., 2020), but includes a regularization term that yields smaller prediction sets; and (3) SAPS (Huang et al., 2024), penalizes the maximal softmax entry according to the rank of $y$. Detailed definitions of these scores are provided in Appendix B.2. We set $\alpha = 0.1$, corresponding to a target coverage of 90%, as common in the literature.

**Details of the CP methods evaluated.** For each score function, we evaluate the following versions.

- Standard: The CP algorithm with the original score function and no modifications.

- Clustered (Ding et al., 2023): The algorithm extends and improves the efficiency of class-wise Mondrian CP (Vovk, 2012) (which applies CP separately to each class) by grouping classes into $M$ clusters based on the similarity of score distributions, and applying CP on each cluster.

  The algorithm has two parameters: $\gamma$, which controls the proportion of data used for clustering, and $M$, the number of clusters. We set $\gamma = 0.2$ to match the proportion used in our methods. $M$ is chosen following (Ding et al., 2023). Further details can be found in Appendix B.4 of their paper.

- AIR (Accumulating Inference Rule): Inspired by the *Climbing Inference Rule* (Goren et al., 2024), which climbs the hierarchy from a predicted leaf node to its parent until reaching a conformalized threshold that guarantees coverage. This approach does not suit the two-level superclass structure of CIFAR-100 and Living-17, as it often leads to the inclusion of all classes. To address this, *we develop an improved variant*. Instead of climbing to the parent, it accumulates mass onto the next superclass with the highest probability, effectively applying conformal prediction at the superclass level rather than the class level.

- MA-CS (Model-Agnostic Class-Similarity): The standard CP algorithm augmented with our binary regularization term, as described in Section 4.

  To select the regularization parameter $\lambda$, we split the calibration set into two equal size sets: $\hat{q}$-calibration (used to compute $\hat{q}$), and $\lambda$-evaluation (used to evaluate performance for different $\lambda$ values). We then iterate over a predefined set of $\lambda$ values and choose the one that achieves the best performance on the $\lambda$-evaluation set.

- MS-CS (Model-Specific Class-Similarity): The standard CP algorithm combined with our regularization term based on the model-specific similarity matrix, as detailed in Section 5. The regularization parameter $\lambda$ is set using the same procedure as in MA-CS.

Note that AIR and our MA-CS cannot be applied for Mini-ImageNet, which lacks a pre-specified superclass structure. On the other hand, our MS-CS is still applicable.

**Evaluation metrics.** The evaluation metrics that we use are the average prediction set size, and for CIFAR-100 and Living-17 also the average number of superclasses in the prediction set. Note that for these metrics: *the lower the better*. The metrics are computed over the test set and we report their means and standard deviations based on 100 trials (random splits of 20% calibration set and 80% test set). We also compute the marginal coverage. The definitions of the metrics are stated in Appendix B.3.

**Results.**

We begin with reporting the top-1 accuracy of each of the four dataset-model pairs: ResNet50 on Mini-ImageNet: 80.38%; ResNet50 on CIFAR-100: 80.93%; ResNet34 on CIFAR-100: 78.92%; ResNet50 on Living-17: 84.68%.

In Table 2 in the appendix, we report the marginal coverage of our MA-CS and MS-CS methods. The pre-specified coverage level is preserved. As expected, our proposed regularization does not

Table 1: Performance comparison of various CP methods with $\alpha = 0.1$.

| Method | #Superclasses ↓ | | | Size ↓ | | | |
|---|---|---|---|---|---|---|---|
| | CIFAR100, RN50 | CIFAR100, RN34 | L17, RN50 | Mini-ImageNet | CIFAR100, RN50 | CIFAR100, RN34 | L17, RN50 |
| **LAC** | | | | | | | |
| Standard | 1.37 (±0.057) | 1.46 (±0.085) | 1.07 (±0.016) | 1.77 (±0.176) | 1.63 (±0.104) | 1.78 (±0.155) | 1.20 (±0.043) |
| Clustered | 1.48 (±0.144) | 1.53 (±0.128) | 1.09 (±0.036) | 2.18 (±0.379) | 1.82 (±0.267) | 1.90 (±0.221) | 1.27 (±0.083) |
| AIR | **1.02** (±0.097) | **1.09** (±0.091) | 1.06 (±0.040) | N/A | 5.10 (±0.101) | 5.45 (±0.089) | 5.30 (±0.061) |
| MA-CS | 1.24 (±0.063) | 1.34 (±0.065) | **1.04** (±0.018) | N/A | 1.54 (±0.127) | 1.70 (±0.120) | 1.19 (±0.037) |
| MS-CS | 1.25 (±0.053) | 1.32 (±0.072) | 1.05 (±0.015) | **1.69** (±0.158) | **1.53** (±0.092) | **1.65** (±0.138) | **1.18** (±0.042) |
| **RAPS** | | | | | | | |
| Standard | 1.91 (±0.066) | 2.69 (±0.102) | 1.19 (±0.025) | 3.84 (±0.186) | 2.70 (±0.128) | 4.34 (±0.208) | 1.51 (±0.056) |
| Clustered | 1.91 (±0.087) | 2.64 (±0.108) | 1.20 (±0.098) | 4.28 (±0.661) | 2.69 (±0.166) | 4.22 (±0.222) | 1.54 (±0.122) |
| AIR | 1.52 (±0.77) | 1.63 (±0.085) | 1.10 (±0.026) | N/A | 7.60 (±0.121) | 8.15 (±0.095) | 5.50 (±0.051) |
| MA-CS | **1.34** (±0.099) | **1.45** (±0.100) | **1.03** (±0.032) | N/A | 2.10 (±0.177) | 2.60 (±0.165) | 1.38 (±0.053) |
| MS-CS | **1.34** (±0.085) | 1.49 (±0.080) | 1.07 (±0.020) | **2.05** (±0.203) | **1.89** (±0.160) | **2.18** (±0.174) | **1.28** (±0.056) |
| **SAPS** | | | | | | | |
| Standard | 1.48 (±0.110) | 1.57 (±0.096) | 1.10 (±0.017) | 2.16 (±0.395) | 1.83 (±0.204) | 1.97 (±0.186) | 1.29 (±0.044) |
| Clustered | 1.60 (±0.265) | 1.73 (±0.299) | 1.20 (±0.224) | 2.96 (±0.760) | 1.99 (±0.336) | 2.18 (±0.406) | 1.49 (±0.236) |
| AIR | **1.16** (±0.039) | **1.10** (±0.023) | **1.00** (±0.017) | N/A | 5.80 (±0.207) | 5.50 (±0.155) | 4.00 (±0.145) |
| MA-CS | 1.26 (±0.044) | 1.42 (±0.063) | 1.06 (±0.012) | N/A | 1.74 (±0.140) | 1.89 (±0.163) | 1.27 (±0.062) |
| MS-CS | 1.36 (±0.084) | 1.39 (±0.059) | 1.07 (±0.013) | **1.95** (±0.239) | **1.71** (±0.172) | **1.81** (±0.136) | **1.24** (±0.049) |

affect this property, which is consistent with the CP theoretical guarantee. In the appendix we report the marginal coverage of the other methods, which also satisfy the specified level.

In Table 1 we report the results for the average prediction set size and, when relevant, the average number of superclasses in the prediction set. Let us discuss these results.

**Comparison between our regularization methods and Standard/Clustered.** Excluding AIR (discussed separately), our methods—MA-CS and MS-CS—consistently achieve the best performance on both metrics across all dataset–model pairs and all CP methods. For example, on CIFAR100–ResNet50 on RAPS score, MA-CS and MS-CS obtain average set size of 1.34, compared to 1.92 for Standard and Clustered, representing a reduction of more than 30%. Similarly, they achieve #Superclasses values of 1.89 and 2.10, whereas Standard and Clustered yield 2.70 and 2.69, corresponding to reductions of 30% and approximately 23%, respectively.

**Comparison between our regularization methods and AIR.** For the #Superclasses metric, performance varies across score functions: under RAPS, our methods outperform AIR, whereas under LAC and SAPS, AIR achieves lower values. For example, on CIFAR100–ResNet34 with the RAPS score, MA-CS and MS-CS achieve #Superclasses values of 1.45 and 1.49, compared to 1.63 for AIR. In contrast, on the same dataset with the SAPS score, MA-CS and MS-CS achieve 1.42 and 1.39, while AIR achieves 1.10.

Importantly, for the average size metric, our methods consistently and significantly outperform AIR across all settings. For instance, on L17–ResNet50 with the RAPS score, MA-CS and MS-CS achieve values of 1.19 and 1.18, compared to 5.30 for AIR, corresponding to a substantial reduction of approximately 78%. Similar reductions are observed throughout the remaining results.

**Comparison between MA-CS and MS-CS.** Although the two methods achieve similar overall performance, MS is slightly better in most settings. The improvement in the prediction set size can be attributed to the use of smaller groups and the higher flexibility of MS, which leverages model-specific information. Interestingly, for #Superclasses the results are similar to those of MA, despite the distances between classes being derived directly from the model.

# 7 CONCLUSION

In this paper, we proposed a class-similarity-based regularization approach that can be applied to any CP score function and reduces *both* the number of groups (e.g., superclasses) *and* the overall size of the prediction sets. We backed our model-agnostic variant with comprehensive theory, which also motivated us to extend it to a novel model-specific approach. Importantly, the latter reduces the prediction set size even further and does not require any known class structure, making it a widely applicable tool in the CP toolbox

*Remark.* In this paper, we used LLMs to polish writing.

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

## A PROOFS AND ADDITIONAL DERIVATIONS

### A.1 PROOF OF THEOREM 4.5

From the definition of $s_\lambda(x, y)$, for any fixed $x$, the size of the penalized conformal set can be written as

$$|\mathcal{C}_\lambda(x)| = \sum_{y \in \mathcal{Y}_0(x)} \mathbb{I}\{s(x, y) \leq q_\lambda\} + \sum_{y \in \mathcal{Y}_1(x)} \mathbb{I}\{s(x, y) \leq q_\lambda - \lambda\}$$

$$= n_0(x)\hat{F}_0^x(q_\lambda) + n_1(x)\hat{F}_1^x(q_\lambda - \lambda),$$

where in the first equation we used $q_\lambda$ due to Assumption 1, and in the second equation we used the definition of $\hat{F}_z^x(t)$. By the law of total expectation,

$$\mathbb{E}[|\mathcal{C}_\lambda(X)|] = \mathbb{E}[n_0(X)\hat{F}_0^X(q_\lambda)] + \mathbb{E}[n_1(X)\hat{F}_1^X(q_\lambda - \lambda)].$$

Using the definition of $\tilde{F}_z(t)$ in Assumption 3, we get

$$\mathbb{E}[|\mathcal{C}_\lambda(X)|] = \overline{n}_0\,\tilde{F}_0(q_\lambda) + \overline{n}_1\,\tilde{F}_1(q_\lambda - \lambda). \tag{8}$$

Next, recall that $q_\lambda$ is defined as the $(1 - \alpha)$–quantile of the CDF $F_\lambda(t) := \mathbb{P}(s_\lambda(X, Y) \leq t)$. Namely, $F_\lambda(q_\lambda) = 1 - \alpha$. Observe that

$$F_\lambda(t) = \mathbb{P}(Y \in \mathcal{Y}_0(x))F_0(t) + \mathbb{P}(Y \in \mathcal{Y}_1(x))F_1(t - \lambda)$$

$$= p_0 F_0(t) + p_1 F_1(t - \lambda).$$

Applying implicit differentiation, by differentiating both sides of $F_\lambda(q_\lambda) = 1 - \alpha$ with respect to $\lambda$ (valid due to Assumption 2), we get

$$\frac{\partial F_\lambda}{\partial \lambda}(q_\lambda) + \frac{\partial F_\lambda}{\partial t}(q_\lambda)\frac{dq_\lambda}{d\lambda} = 0.$$

Since

$$\frac{\partial F_\lambda}{\partial t}(t) = p_0 f_0(t) + p_1 f_1(t - \lambda), \quad \frac{\partial F_\lambda}{\partial \lambda}(t) = -p_1 f_1(t - \lambda),$$

we obtain

$$\frac{dq_\lambda}{d\lambda} = \frac{p_1 f_1(q_\lambda - \lambda)}{p_0 f_0(q_\lambda) + p_1 f_1(q_\lambda - \lambda)}.$$

At $\lambda = 0$ (with $q = q_0$),

$$\left.\frac{dq_\lambda}{d\lambda}\right|_{\lambda=0} = \frac{p_1 f_1(q)}{p_0 f_0(q) + p_1 f_1(q)}. \tag{9}$$

Now, let us differentiate equation 8 with respect to $\lambda$ (valid due to Assumption 3):

$$\frac{d}{d\lambda}\mathbb{E}[|\mathcal{C}_\lambda(X)|] = \overline{n}_0\tilde{f}_0(q_\lambda)\frac{dq_\lambda}{d\lambda} + \overline{n}_1\tilde{f}_1(q_\lambda - \lambda)\left(\frac{dq_\lambda}{d\lambda} - 1\right).$$

Evaluating at $\lambda = 0$ and substituting equation 9,

$$\left.\frac{d}{d\lambda}\mathbb{E}[|\mathcal{C}_\lambda(X)|]\right|_{\lambda=0} = \overline{n}_0\tilde{f}_0(q)\frac{p_1 f_1(q)}{p_0 f_0(q) + p_1 f_1(q)} + \overline{n}_1\tilde{f}_1(q)\left(\frac{p_1 f_1(q)}{p_0 f_0(q) + p_1 f_1(q)} - 1\right)$$

$$= \frac{1}{p_0 f_0(q) + p_1 f_1(q)}\left(\tilde{f}_0(q)f_1(q) \cdot p_1\overline{n}_0 - \tilde{f}_1(q)f_0(q) \cdot p_0\overline{n}_1\right).$$

Since $\dfrac{1}{p_0 f_0(q) + p_1 f_1(q)} > 0$ (strictly positive), we obtain equation 3.

## A.2 Derivation of equation 4

To simplify the notation, define the event $A_z = Y \in \mathcal{Y}_z(X)$. We have

$$
\begin{aligned}
F_z(t) &= \mathbb{P}(s(X,Y) \leq t | A_z) \\
&= \mathbb{E}_{X,Y|A_z}[\mathbb{I}\{s(X,Y) \leq t\}] \\
&= \mathbb{E}_{X|A_z}[\mathbb{E}_{Y|A_z,X}[\mathbb{I}\{s(X,Y) \leq t\}]] \\
&= \mathbb{E}_{X|A_z}[\mathbb{P}(s(x,Y) \leq t | Y \in \mathcal{Y}_z(x), X = x)] \\
&= \mathbb{E}_{X|A_z}[F_z^X(t)]
\end{aligned}
$$

Next, using $p_{X|A_z}(x) = \dfrac{\mathbb{P}(A_z | X = x)p_X(x)}{\mathbb{P}(A_z)} = \dfrac{p_z(x)p_X(x)}{p_z}$, we have

$$
\begin{aligned}
F_z(t) &= \int F_z^x(t)p_{X|A_z}(x)dx \\
&= \frac{1}{p_z}\int F_z^x(t)p_z(x)p_X(x)dx \\
&= \frac{1}{p_z}\mathbb{E}[p_z(X)F_z^X(t)].
\end{aligned}
$$

# B Additional Experimental Details

## B.1 Training details

For Cifar-100 models, we use the following: Batch size: 128; Epochs: 100; Cross entropy loss; Optimizer: SGD; Learning rate: 0.1; Momentum 0.9 and weight decay 0.0005.
Similarly, for Living 17 we use: Batch size: 64; Epochs: 15; Cross entropy loss; Optimizer: Adam; Learning rate: 0.0001.
For training details regarding Mini-ImageNet, see the following link:
https://huggingface.co/datasets/timm/mini-ImageNet

## B.2 Definitions of the score functions

**LAC**:

$$
s(x,y) := 1 - \hat{\pi}(x,y) \tag{10}
$$

**RAPS**:

$$
s(x,y) := \sum_{y'=1}^{C} \hat{\pi}(x,y')\,\mathbf{1}\{\hat{\pi}(x,y') > \hat{\pi}(x,y)\} + \lambda_1 \cdot \left(o_x(y) - k_{\text{reg}}\right)^+ + \hat{\pi}(x,y) \cdot u, \tag{11}
$$

where

$$
o_x(y) = \left| \{\, y' \in \mathcal{Y} : \hat{\pi}(x,y') \geq \hat{\pi}(x,y) \,\} \right|
$$

, $(x)^+$ is the positive part of the expression and $\lambda_1, k_{reg}$ are the hyperparameters of RAPS.

**SAPS**:

$$
S(x,y) := \begin{cases} u \cdot \hat{\pi}_{\max}(x,y), & \text{if } o_x(y) = 1 \\ \hat{\pi}_{\max}(x,y) + \left(o_x(y) - 2 + u\right) \cdot \lambda_1, & \text{else} \end{cases} \tag{12}
$$

where $u$ is a uniform random variable and $\hat{\pi}_{\max}(x,y)$ denotes the maximum softmax.

## B.3 Definitions of the evaluation metrics

We report metrics over the test set, which we denote by $\{(\mathbf{x}_i^{(test)}, y_i^{(test)})\}_{i=1}^{N_{test}}$, comprising of the samples that were not included in the calibration set or CP set. The metrics are as follows.

- *Average set size* (AvgSize) – The mean prediction set size of the CP algorithm:

$$\text{AvgSize} = \frac{1}{N_{test}} \sum_{i=1}^{N_{test}} |\mathcal{C}(\mathbf{x}_i^{(test)})|.$$

- *Average number of superclasses* (AvgSC) - The mean number of distinct superclasses in prediction set of the CP algorithm.

$$\text{AvgSC} = \frac{1}{N_{test}} \sum_{i=1}^{N_{test}} |\mathcal{G}(\mathbf{x}_i^{(test)})|.$$

where $\mathcal{G}(x) = \{g(y) : y \in \mathcal{C}(x)\}$

- *marginal coverage* - The coverage rate guarantee in prediction sets of the CP algorithm:

$$\text{Coverage rate} = \frac{1}{N_{test}} \sum_{i=1}^{N_{test}} \mathbf{1}\{y_i \in \mathcal{C}(x_i^{(test)})\}.$$

## C    ADDITIONAL EXPERIMENTS

**Metrics as function of $\lambda$.**

We further investigate the effect of the regularization parameter $\lambda$ by plotting AvgSize and AvgSC as functions of $\lambda$ for the MA method using the RAPS score. The experiments are conducted on the CIFAR100–ResNet50 dataset–model pair. From Figure 2, we observe that there exists a value of $\lambda$ that minimizes AvgSize. Figure 3 shows that AvgSC decreases as $\lambda$ increases.

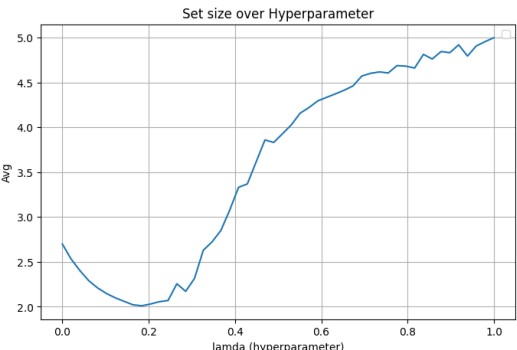

Figure 2: Fine-tuning hyperparamter $\lambda$ in RAPS method Model Agnostic with model Cifar-100 Resnet 50. We can see that a range of $\lambda$ values brings the set size to minimum over the baseline.

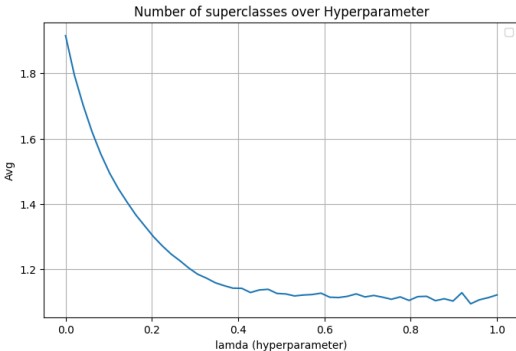

Figure 3: number of superclasses over hyperparamter $\lambda$ in RAPS method Model Agnostic with model Cifar-100 Resnet 50. Similarly, the number of superclasses is decreasing along with $\lambda$.

**Marginal coverage results.**

In Table 2, we present the marginal coverage for each of the methods for all settings. As expected from Theorem 3.1, the marginal coverage holds.

Table 2: Marginal coverage of the CP methods for $\alpha = 0.1$.

| | Coverage | | | |
|---|---|---|---|---|
| Method | Mini ImageNet | CIFAR100 RN50 | CIFAR100 RN34 | L17 |
| **LAC** | | | | |
| Standard | 0.901 | 0.899 | 0.902 | 0.905 |
| Clustered | 0.895 | 0.915 | 0.919 | 0.902 |
| AIR | 0.898 | 0.905 | 0.902 | 0.901 |
| MA-CS | N/A | 0.9 | 0.895 | 0.898 |
| MS-CS | 0.896 | 0.897 | 0.899 | 0.899 |
| **RAPS** | | | | |
| Standard | 0.898 | 0.9 | 0.903 | 0.902 |
| Clustered | 0.902 | 0.917 | 0.913 | 0.903 |
| AIR | 0.897 | 0.906 | 0.907 | 0.904 |
| MA-CS | N/A | 0.899 | 0.899 | 0.9 |
| MS-CS | 0.901 | 0.898 | 0.9 | 0.905 |
| **SAPS** | | | | |
| Standard | 0.9 | 0.898 | 0.904 | 0.899 |
| Clustered | 0.901 | 0.9 | 0.9 | 0.902 |
| AIR | 0.903 | 0.908 | 0.896 | 0.900 |
| MA-CS | N/A | 0.898 | 0.9 | 0.898 |
| MS-CS | 0.898 | 0.9 | 0.899 | 0.9 |

