# OpenReview forum: "Leveraging Class Similarity for Enhanced Conformal Prediction"
_ICLR.cc/2026/Conference — ICLR 2026 Conference Withdrawn Submission_

### Official Review · Reviewer_c9AR · 2025-10-24

**Soundness:** 2
**Presentation:** 2
**Contribution:** 3
**Rating:** 2
**Confidence:** 2

**Summary:**

This paper introduces a new framework for improving Conformal Prediction (CP) in classification tasks by incorporating class similarity information into the CP score function. Traditional CP guarantees marginal coverage but does not account for the semantic coherence of labels within the prediction set. Thus, the authors propose Model-Agnostic Class Similarity (MA-CS)  and Model-Specific Class Similarity (MS-CS).

**Strengths:**

1. The paper goals to ensure semantic coherence in CP prediction sets, which is a real gap in CP's applicability to high-stakes domains.

2. MS-CS moves from human-defined partitions to model-driven class similarity is creative.

3. The experiments include three datasets (CIFAR-100, Living-17, Mini-ImageNet) and three score functions (LAC, RAPS, SAPS).

**Weaknesses:**

1. For the MA-CS variant, performance may depend heavily on the quality of predefined groupings and similarity measurement functions. Besides, the "quality" of similariy also depends on the performance of the model.

Similarly, both methods rely on a regularization parameter $\lambda$. How sensitive are the results to the choice of $\lambda$? How does cosine similarity compare (empirically or theoretically) to other kernel-based similarities?

2. The theoretical results assume *well-calibrated softmax outputs* and *consistent class groupings*.
It would be better to include experiments or analysis under unexpected cases, such as noisy labels, unreliable (predefined or learned) group partitions, or poor model performance.

How does performance degrade if the predefined groups are noisy or partially incorrect?

3. Some theoretical sections (especially Theorem 4.5 and its discussion) are dense and could benefit from additional intuition or graphical illustrations.

4. Could the framework generalize to other data formats, such as text or tabular? If yes, it is recommended to show more results.
Furthermore, if it fits structured prediction or multi-label settings where class relationships are hierarchical or overlapping?

5. The writing/presentation is hard to follow, making the manuscript "incomplete".

**Questions:**

See above.

---

### Official Review · Reviewer_JwSg · 2025-10-28

**Soundness:** 2
**Presentation:** 3
**Contribution:** 2
**Rating:** 2
**Confidence:** 3

**Summary:**

The paper proposes two new conformal score functions (more precisely, two ways of modifying any existing score function) that achieve two goals. The primary goal is to promote semantic similarity of labels in the resulting prediction set (e.g., classes that belong to the same group). The secondary goal is to reduce the average set size.  The proposed score functions are obtained by combining an existing score function with an additive penalty on the distance $d$ between the input $y$ and the model’s top-1 prediction $\hat{y}$. The first score function sets $d(y, \hat{y})$ to be the indicator for whether $y$ and $\hat{y}$ are in the same group (where groups are pre-defined). The second score function sets $d(y, \hat{y})$ to be the distance between the centroid of embedded examples of class $y$ and embedded examples of $\hat{y}$, where the embedding is the output of the penultimate layer of the neural network (before the regression head).

**Strengths:**

* The paper is generally well written.
* The idea of using the NN’s learned features to measure class similarity (as described in Sec 5) is nice.

**Weaknesses:**

* Unclear practical relevance. There are two reasons for this:
    * First, I am not convinced by the problem motivation (as described in the second paragraph of the introduction). Ambiguity can be useful for determining when further investigation or caution is needed. In the autonomous driving example, let’s consider a rare event that requires an uncommon response: the car is currently going at high speed, and a pedestrian has stepped out into the road in front of the car. In this case, we want the car to honk, so the pedestrian can get out of the way. However, using your sets, if the model’s top-1 prediction is not “pedestrian crossing”, then your method makes it so the prediction set will not contain “pedestrian crossing” in most cases.
    * Second, the empirical results are a bit weak, in my reading. Standard with LAC is a strong baseline, and differences between the proposed methods and this baseline do not generally seem to be statistically significant.
* No consideration of class-conditional coverage (see Question below)
* The theory is not particularly surprising
* Assumption 1 is not stated very clearly. It would be useful to write out the mathematical definition of the “statistical quantile.” Does this just mean that we replace the score threshold with the population quantile?

**Questions:**

* How do MA-CS and MS-CS impact class-conditional coverage? Can you compute the average class-conditional coverage gap of the baselines and your methods?
* How does this paper relate to https://arxiv.org/pdf/2410.01767 (Cortes-Gomez et al., 2025)? The setting in that paper seems quite related.
* On line 418 there is a comment about how one of the baselines and one of the proposed methods cannot be applied to Mini-ImageNet due to the lack of a superclass structure. Why can the WordNet hierarchy that exists for full ImageNet not be used here? Moreover, why are the experiments run on Mini-ImageNet instead of the full ImageNet? All of the experiments are run datasets with roughly 100 classes or less. Would it not work on the 1000 classes in full ImageNet?
* The problem the paper solves is “how do we encourage prediction sets to contain labels from a single semantic group?” and not “how do we encourage prediction sets to contain labels from as few semantic groups as possible?” With strong classifiers (with, e.g., 80% top-1 accuracy, as is the case in the experiments), these problems are effectively equivalent. But in general, these problems are not the same. The score function $s_{\lambda}(x,y) = s(x,y) + \lambda d(y, \hat y(x))$ (Eq. 2) does not capture the fact that if we want to solve the second question, once we decide to include a an out-group class, it is better to include classes from the same group as that class than a class from a not-yet-included class.
    * First, a suggestion: It could be useful to highlight the difference between these two problems. Currently, the language in the abstract (“users can benefit from prediction sets … [that] contain a small number of semantically different groups”) led me to initially believe that the paper would address the second problem
    * Second, a question: Do you have ideas about how your approach can be adapted to solve the second problem?
* Sadinle et al, 2019 says Standard CP with the LAC score approximates the theoretically optimal smallest sets. How should readers interpret your Theorem 4.5 in the context of this existing optimal size result?

Small comment:
* Line 167: “semantical” -> semantic

---

### Official Review · Reviewer_xAqH · 2025-10-28

**Soundness:** 1
**Presentation:** 2
**Contribution:** 1
**Rating:** 2
**Confidence:** 5

**Summary:**

This paper modifies the score function by adding a label similarity.

**Strengths:**

The empirical result is good.

**Weaknesses:**

1. The contribution is very mild. Adding a similarity to the score function is clearly not enough for publication at ICLR. I don’t have much more to say. It is a clear rejection for me.

2. If the similarity is helpful for prediction, it shouldn’t only appear in the conformal prediction set size; it should appear in other accuracy measurements as well. Even if the similarity is helpful for classification, this paper makes no contribution to conformal prediction. Adding a useful measure to the score function can clearly help improve conformal prediction results.

**Questions:**

None.

---

### Official Review · Reviewer_JRi9 · 2025-10-31

**Soundness:** 3
**Presentation:** 3
**Contribution:** 2
**Rating:** 4
**Confidence:** 3

**Summary:**

The paper proposes a regularization framework for Conformal Prediction (CP) that leverages class similarity to improve the efficiency and semantic similarity of prediction sets. In the model-agnostic version (MA-CS), a binary distance function penalizes classes belonging to different predefined groups, preserving exchangeability and theoretical coverage guarantees while reducing average set size. The model-specific variant (MS-CS) replaces the predefined grouping with a similarity measure derived from the cosine similarity of centered class mean feature embeddings of a NNs (no need for human-defined groups). Experiments on four image datasets and two ResNet models show that both variants achieve the target coverage (with minor variability) while producing smaller and more coherent prediction sets than regularization baselines.

**Strengths:**

* The paper is clearly organized and effectively motivates the problem. It addresses the lack of semantic coherence in conformal prediction sets, enhancing their practical usefulness for downstream tasks such as healthcare or autonomous systems.

* The proposed regularization approach is conceptually simple, easy to implement, and can be integrated with existing nonconformity scores and CP frameworks.

* For the model-agnostic variant (MA-CS), the authors provide a formal analysis demonstrating that coverage is preserved and the average set size can be reduced. The extension to model specific class similarity sounds reasonable and well-motivated.

* The experimental results are promising for both MA-CS and MS-CS, showing improvements in comparison with other regularization strategies across four image datasets and two network architectures.

**Weaknesses:**

1. The model-specific variant (MS-CS), while well motivated, lacks a more formal analysis. It remains unclear what the cosine-based similarity matrix is truly capturing and how it aligns with meaningful semantic concepts/groupings. An examination of these learned relationships would provide valuable insight.

2. The method’s performance depends on the regularization weight λ but the paper offers limited discussion on its robustness. Since λ is chosen based on splitting the calibration set (Paragraph in L408), it would be useful to analyze how these choices affects the variance of the estimated quantile and the resulting coverage.

3. It is not discussed how the approach behaves under different target error rates (values of α other than 0.1).

4. Although the empirical improvements are consistent, they can be modest in some settings, raising questions about when and why the method yields the largest benefits. Is there some general characteristic that we can identify here? Under which conditions do this approach make a difference?

5. Experiments are restricted to image classification with resnet models; evaluating the approach on other modalities (text, tabular data) would better support claims of generality from an empirical point of view.

**Questions:**

Please, see weaknesses.

---

### Note · Authors · 2025-11-14

**Comment:**

We sincerely thank the reviewers for the time and effort they dedicated to evaluating our submission. After careful consideration, we have decided to withdraw the paper.

We feel that our practical and theoretical contributions were not communicated as clearly as we intended. For example, our work demonstrates the ability to improve both prediction-set sizes and the number of superclasses by tuning a single hyperparameter in just a few minutes using only a small portion of the data and our theory explains it. Additionally, we highlight the consistent improvement of the LAC algorithm, which, to the best of our knowledge, has not been reported previously.

Nevertheless, we greatly appreciate the reviewers’ feedback and will carefully consider it as we work to strengthen the clarity, quality, and impact of our work in a future submission.

**Withdrawal Confirmation:**

I have read and agree with the venue's withdrawal policy on behalf of myself and my co-authors.